# Quantitative analysis of rabies virus-based synaptic connectivity tracing

**Alexandra Tran-Van-Minh**[1][☉]**, Zhiwen Ye**[ID][1][☉]**, Ede Rancz**[ID][1,2]*

1 The Francis Crick Institute, London, United Kingdom, 2 INMED, INSERM, Aix-Marseille Université, Marseille, France

☉ These authors contributed equally to this work.
* ede.rancz@inserm.fr

**Data Availability Statement:** Cell counts for all experiments and the code used for analysis and generating figures can be found at GitHub (https://github.com/ranczlab/).

**Funding:** This work was supported by the Wellcome Trust (104285/B/14/Z) and the Francis

## Abstract

Monosynaptically restricted rabies viruses have been used for more than a decade for synaptic connectivity tracing. However, the verisimilitude of quantitative conclusions drawn from these experiments is largely unknown. The primary reason is the simple metrics commonly used, which generally disregard the effect of starter cell numbers. Here we present an experimental dataset with a broad range of starter cell numbers and explore their relationship with the number of input cells across the brain using descriptive statistics and modelling. We show that starter cell numbers strongly affect input fraction and convergence index measures, making quantitative comparisons unreliable. Furthermore, we suggest a principled way to analyse rabies derived connectivity data by taking advantage of the starter vs input cell relationship that we describe and validate across independent datasets.

## 1 Introduction

Understanding synaptic connectivity is important to unravel the workings of the nervous system. Recently developed variants of modified rabies virus provide a powerful tool to determine the upstream connectivity of neuronal populations or even single neurons [1–4] at the level of the whole brain, a feat currently unattainable by other means. Monosynaptically restricted connectivity tracing [5, 6] can be initiated in neuronal populations defined in various ways, for example genetically [7], functionally [8] or by projection targets [9]. This method offers a highly quantifiable measure with single-cell resolution, i.e. individual rabies infected neurons. When combined with whole-brain imaging and segmentation, this approach can reveal high-fidelity input connectivity maps. It is hard however to precisely control the number of starter cells labelled by initial virus injection. This source of variability is usually ignored, as it is often assumed that the brain-wide number of input cells ($n_i$) scales linearly with increasing number of labelled starter cells ($n_s$) [10–15]. Although it is a reasonable assumption that $n_i$ increases monotonically with $n_s$, a linear relationship between $n_i$ and $n_s$ would require that labelled starter cells do not share any presynaptic input cell, and that the pool of input cells is infinite. These assumptions are biologically implausible, and the exact relationship between $n_i$ and $n_s$ remains largely unexplored. This is in part because most rabies tracing studies are performed on small numbers of brains, starter cells are not quantified, or input cells may have been

Crick Institute, which receives its core funding from Cancer Research UK (FC001153), the United Kingdom Medical Research Council (FC001153), and the Wellcome Trust (FC001153). A.TVM. received funding from the European Union's Horizon 2020 research and innovation programme under the Marie Sklodowska-Curie grant agreement No 747902. The funders had no role in study design, data collection and analysis, decision to publish, or preparation of the manuscript.

**Competing interests:** The authors have declared that no competing interests exist.

counted only in selected parts of the brain. Indeed, the advent of automated pipelines for registration and cell counting in whole-brain images is relatively recent [14, 16, 17], and without these pipelines, whole-brain cell counting for large datasets was an extremely time- and labour-intensive process. In addition, various methods are used to analyze rabies tracing experiments. Metrics used in previous studies include input fraction (the inputs count per area relative to either the total number of input cells in the brain, sometimes referred to as total input fraction [15, 18–21], or relative to the number of input cells in a given area [15, 22] or layer [23]), and the convergence index, also called input connection strength index or input magnitude (area inputs count normalized by number of starter cells [18, 20, 24–26]). The disparity in methods and terminology used to report the properties of connectivity maps complicates direct comparisons across various studies.

Here we set out to explore the relationship between the number of starter and input cells. We generated a comparatively large dataset with a broad range of starter cell numbers in layer 5 of mouse visual cortex. We then used descriptive statistics, model selection and numerical modelling of connectivity to explore the data, and compare various analysis approaches. We find that the relationship between number of the starter cells and number of the labelled input cells is non-linear, and that the range of starter cells differently affects some of the metrics commonly used to analyze connectivity maps. Using these results we suggest principles for experiment design and appropriate analysis methodologies to use depending on the data available and on the desired information to be extracted.

## 2 Results

We carried out rabies tracing experiments initiated in layer 5 pyramidal neurons (L5PN) in mouse primary and secondary visual cortex. We followed previously described protocols [3] using the G-protein deleted, EnvA pseudotyped version of CVS-N2c rabies virus [6]. To achieve a wide range of starter cell numbers, we used different labeling strategies. We took advantage of varied cell densities in Cre driver lines labeling L5PNs, or through retrogradely labeling L5PNs by injecting retrograde AAV viruses to their projection targets. In addition, we also varied the volume of the injected helper viruses. Our dataset is thus uniform in terms of cell-type (L5PN), yet contains sufficient heterogeneity in terms of sub-types and starter areas to test comparative approaches. Whole brains were imaged using 2-photon tomography, then registered, segmented and annotated according to the Allen Common Coordinate Framework (CCFv3, [27]) using the open source brainreg software [17]. Starter and input cells were detected and classified using the open source software cellfinder [28]. Area-wise cell counts for individual experiments are reported in S1 Text.

We use the following nomenclature throughout. Starter cells (also called postsynaptic, host or target cells in the literature) are neurons for which rabies tracing was initiated, and their count is denoted by $n_s$. Neurons labelled after the transsynaptic jump, and thus considered first-order presynaptic to starter cells, are called input cells and their count in the entire brain is denoted by $n_i$. Starter and input areas are brain areas defined according to the CCFv3 containing starter or input cells, respectively. Brain area names and abbreviations are from the Allen Common Coordinate Framework (CCFv3, [27]).

### 2.1 The relationship between input and starter cells is non-linear and is best described in log scale

We took advantage of the broad range of starter cell numbers and the large number of experiments in our dataset to investigate the nature of the $n_i$ vs $n_s$ relationship. First, we fitted whole-brain $n_i$ vs $n_s$ data with linear, quadratic, exponential and power-law functions as well as a

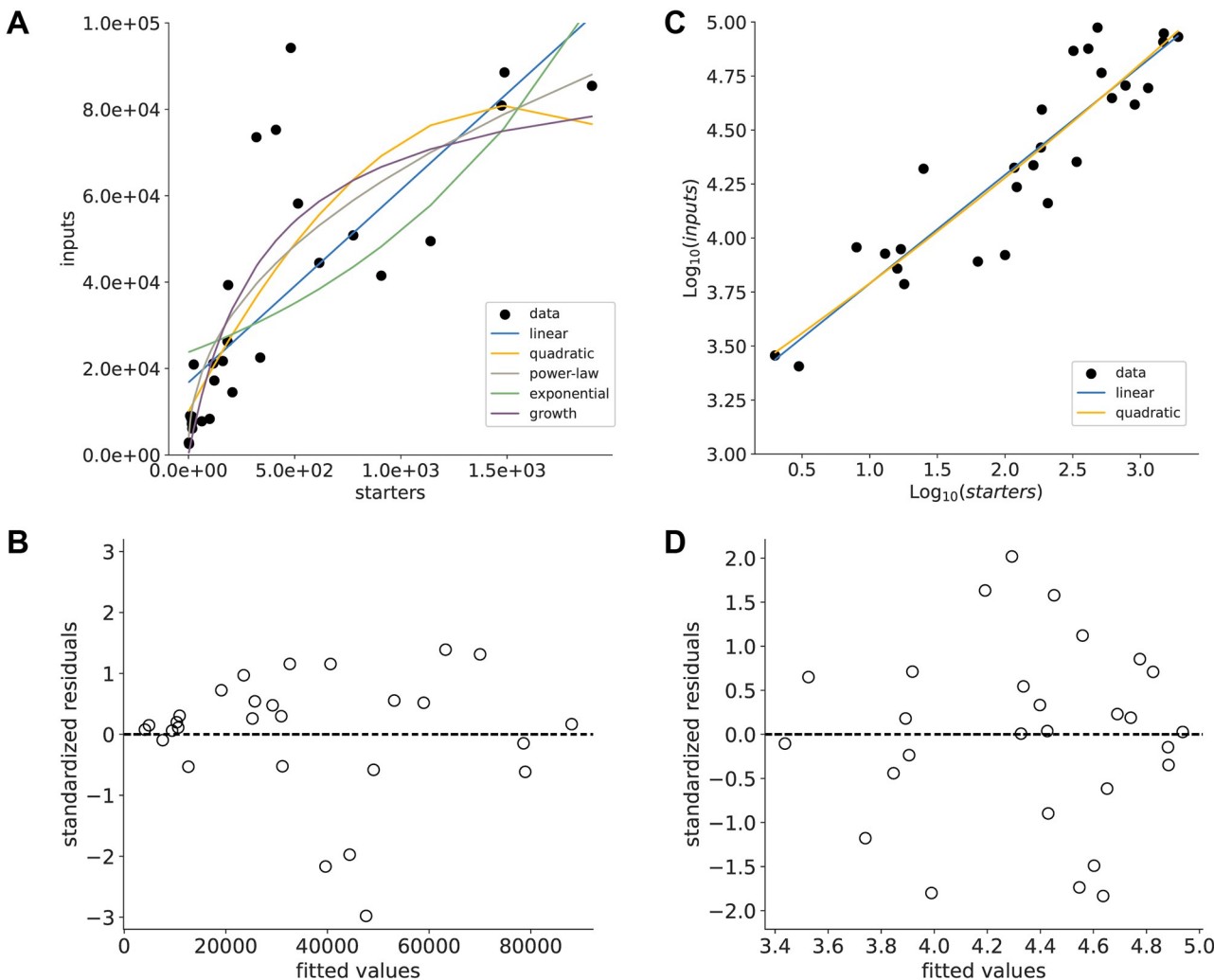

**Fig 1. Model comparison for whole-brain data.** (A) Number of input cells vs starter cells. Colours indicate different fitted models (blue—linear; yellow —quadratic; grey—power law; green—exponential; purple—growth). (B) Distribution of standardized residuals for the power-law fit. (C) Same as in A, but on log-transformed data. (D) Same as in B, but for the linear fit on log-transformed data.

growth model that describes the labeling of input cells at a given rate, and considering a maximum number of input cells that can be labeled (Fig 1A). We then performed model selection using the Akaike Information Criterion corrected for small sample sizes ($AIC_c$) [29]. From the model selection procedure, the power-law model and growth models are the best candidates (Table 1). However, residuals showed strong heteroscedasticity when standardized residuals were plotted against the fitted values (Fig 1B). In such cases, logarithmic transformation may allow to adjust the data distribution to a less skewed, more Gaussian-like distribution [30]. Model selection was next performed on log-transformed data and gave support for the fitting of a linear model, as the addition of a quadratic term did not improve the model fit (Fig 1C, Table 2). The plots of standardized residuals vs fitted values suggest a better normality of residuals in the case of log-transformed data (Fig 1B and 1D). There are thus two possible error models: log-normal distributed errors based on untransformed data, or normally distributed errors based on fitting log-transformed data. We tested these possibilities by fitting two models in which the likelihood form explicitly includes the error structures [31]. The two resulting

**Table 1. ΔAICc for untransformed data.** External datasets are from [13–15, 18, 19, 21, 22, 25, 32–36]. For easier comparison, we show the ΔAICc values, calculated as the difference between the AICc value for each model and the lowest AICc value per dataset. Lower AIC values indicate a better-fit model, and a model with a ΔAIC (the difference between the two AIC values being compared) of more than -2 is considered significantly better than the model it is being compared to.

| dataset | rabies | linear | power-law | exponential | quadratic | growth |
|---|---|---|---|---|---|---|
| this dataset | CVS-N2c | 10.75 | **1.24** | 17.60 | 20.44 | **0.00** |
| Beier [19] | SAD-B19 | 23.66 | 10.00 | 30.93 | 19.97 | **0.00** |
| Brown [22] | SAD-B19 | 4.26 | 2.20 | 4.26 | 12.02 | **0.00** |
| Fu [32] | SAD-B19 | **1.08** | **0.00** | 18.17 | 14.42 | **1.29** |
| Gehrlach [33] | SAD-B19 | 2.31 | **1.04** | 2.63 | 16.54 | **0.00** |
| Graham interneuron [21] | CVS-N2c | **1.15** | **0.55** | 14.79 | 15.74 | **0.00** |
| Graham projection [21] | CVS-N2c | **0.29** | **0.00** | 13.08 | 16.52 | **0.91** |
| Hafner [18] | SAD-B19 | **0.00** | **0.00** | 9.13 | 17.94 | **0.01** |
| Kim [25] | SAD-B19 | 4.00 | **1.94** | 4.75 | 11.53 | **0.00** |
| Pouchelon P30-P42 [13] | CVS-N2c | 2.75 | **1.20** | 5.93 | 14.15 | **0.00** |
| Pouchelon P5-P10 [13] | CVS-N2c | **0.47** | **0.00** | 28.94 | 15.85 | **0.94** |
| Sun [14] | SAD-B19 | 32.32 | 14.48 | **0.00** | 23.80 | 36.58 |
| Takatoh [36] | SAD-B19 | **0.00** | **0.73** | **4.00** | 15.58 | **1.87** |
| Vinograd [34] | SAD-B19 | **1.55** | **0.82** | **1.63** | 17.19 | **0.00** |
| Wee [15] | SAD-B19 | 2.72 | **0.00** | 5.62 | 17.55 | **1.99** |
| Allen Institute [35] | CVS-N2c | 7.16 | **0.00** | 17.17 | 14.32 | 3.58 |

models were then compared using the $AIC_c$ and confirmed support for the log-normal error model (Table 3). Consequently, we used log-transformed $n_i$ and $n_s$ data for further analysis.

To test the generality of our conclusions, we performed model selection and error model analysis on datasets from published studies performed on a range of cell types and starter brain areas, using different imaging and cell counting approaches as well as employing various rabies virus strains and G-proteins [13–15, 18, 19, 22, 25, 32–36]. The analysis revealed statistical support for the power-law and growth models for untransformed $n_i$ vs $n_s$ data, a linear model for the log-transformed data, and for the log-normal error model (S1 Fig, Tables 1–3)

**Table 2. ΔAICc for log-transformed data.**

| dataset | rabies | linear | quadratic |
|---|---|---|---|
| this dataset | CVS-N2c | **0.00** | 15.99 |
| Beier | SAD-B19 | **0.00** | 13.47 |
| Brown | SAD-B19 | **0.00** | 11.97 |
| Fu | SAD-B19 | **0.00** | 14.18 |
| Gehrlach | SAD-B19 | **0.00** | 15.93 |
| Graham interneuron | CVS-N2c | **0.00** | 15.25 |
| Graham projection | CVS-N2c | **0.00** | 16.23 |
| Hafner | SAD-B19 | **0.00** | 17.55 |
| Kim | SAD-B19 | **0.00** | 13.43 |
| Pouchelon—P30-P42 | CVS-N2c | **0.00** | 16.03 |
| Pouchelon—P5-P10 | CVS-N2c | **0.00** | 16.15 |
| Sun | SAD-B19 | **0.00** | 12.90 |
| Takatoh | SAD-B19 | **0.00** | 16.34 |
| Vinograd | SAD-B19 | **0.00** | 16.28 |
| Wee | SAD-B19 | **0.00** | 15.70 |
| Allen Institute | CVS-N2c | **0.00** | 16.15 |

**Table 3. ΔAICc for residuals distribution.**

| dataset | rabies | log-normal | normal |
|---|---|---|---|
| this dataset | CVS-N2c | **0.00** | 25.20 |
| Beier | SAD-B19 | **0.00** | 189.86 |
| Brown | SAD-B19 | **0.00** | 9.84 |
| Fu | SAD-B19 | **0.00** | 7.81 |
| Gehrlach | SAD-B19 | **0.00** | 7.52 |
| Graham interneuron | CVS-N2c | 6.33 | **0.00** |
| Graham projection | CVS-N2c | **0.00** | 2.39 |
| Hafner | SAD-B19 | **0.42** | **0.00** |
| Kim | SAD-B19 | **0.00** | 8.17 |
| Pouchelon P30-P42 | CVS-N2c | **0.00** | 4.36 |
| Pouchelon P5-P10 | CVS-N2c | **0.00** | 15.08 |
| Sun | SAD-B19 | **0.00** | 24.50 |
| Takatoh | SAD-B19 | **0.00** | 2.46 |
| Vinograd | SAD-B19 | **0.00** | 4.66 |
| Wee | SAD-B19 | **0.00** | **1.02** |
| Allen Institute | CVS-N2c | 6.59 | **0.00** |

across most datasets. Therefore, the growth of labelled inputs with the number of starter cells follows broadly the same statistical rules independently of experimental conditions and is not specific to our dataset. Furthermore, this generality allowed us to investigate whether rabies strains may affect the parameters of the $n_i$ vs $n_s$ relationship. We used a linear classifier to determine whether the rabies strain used could be identified using the $\log(n_s)$, $\log(n_i)$ and the broad class of starter cells (pyramidal or interneuron) as features. The classifier could predict the rabies strain with 82% accuracy (S2 Fig) and revealed a general trend where brains labelled using the CVS-N2c rabies strains displayed higher input cell counts for a similar number of starter cells, consistent with a higher trans-synaptic yield of the CVS-N2c strain [6].

We next built a probabilistic connectivity model (S3 Fig) and explored the influence of various model parameters on the structure of the data. This model assumes that for a given input area $I$ containing $N_I$ cells, each input neuron connects with a given starter neuron with probability $p$. This model allows us to simulate a rabies tracing experiment, by selecting a number $n_s$ of starter neurons, and building a simulated connectome of all their connections with area $I$. From each modeled connectome, we can then calculate $n_i$, the total number of unique neurons in $I$ connected to the $n_s$ starter neurons. In order to represent variability across brains, the model parameters (connection probability $p$ and input pool size $N_i$) were sampled from Gaussian distributions for each simulated connectome (S4A Fig). The resulting $n_i$ vs $n_s$ curve displays a strong skewness of residuals, similar to the experimental data (S4B Fig). This process was repeated 100 times with random sampling, and we performed the error model analysis for each resulting $n_i$ vs $n_s$ curve. The distribution of resulting $AIC_c$ values shows a consistent, strong support for the log-normal error model (S4C Fig). Therefore, the probabilistic model, built with minimal assumptions and small number of parameters, is sufficient to generate $n_i$ vs $n_s$ relationships with a log-normal error distribution, similar to that of the experimental data.

## 2.2 Input vs. starter relationships parameters differ across input areas

For further analysis, we have selected 19 functionally diverse input areas (Table 4). To determine whether the number of input cells from individual brain areas displayed similar form and error structure as the whole-brain data, we applied the same model selection procedure

**Table 4. ΔAICc for untransformed data for individual brain areas.**

| dataset | Linear | Power-law | Exponential | Quadratic | growth |
|---|---|---|---|---|---|
| VISp | 5.91 | **0.00** | 16.33 | 16.35 | **0.39** |
| VISpm | 9.10 | 2.98 | 10.92 | 12.26 | **0.00** |
| VISl | 9.17 | **0.61** | 14.60 | 19.52 | **0.00** |
| VISam | 10.99 | 3.11 | 13.30 | 17.15 | **0.00** |
| VISal | 2.36 | **0.00** | 5.41 | 18.25 | 3.14 |
| RSPagl | 10.52 | 3.45 | 13.26 | 18.03 | **0.00** |
| RSPd | 8.90 | 2.07 | 12.44 | 17.95 | **0.00** |
| RSPv | 11.14 | 3.13 | 13.91 | 18.23 | **0.00** |
| AM | 11.57 | 3.96 | 13.27 | 21.18 | **0.00** |
| LD | 13.67 | **0.00** | 18.15 | 20.51 | **0.48** |
| LP | 7.68 | **0.15** | 12.03 | 17.53 | **0.00** |
| LGd | **0.00** | **1.28** | **1.49** | 16.08 | 6.12 |
| ORB | 7.32 | **1.64** | 9.80 | 19.40 | **0.00** |
| ACA | 7.41 | 2.03 | 10.75 | 19.83 | **0.00** |
| MOs | **0.39** | **0.99** | **0.00** | 15.40 | **1.40** |
| CLA | 5.80 | **0.81** | 9.43 | 19.91 | **0.00** |
| PTLp | **1.00** | **0.00** | 2.71 | 17.08 | 3.02 |
| TEa | 7.32 | **0.85** | 15.18 | 19.16 | **0.00** |
| AUD | 5.83 | **0.00** | 17.33 | 18.38 | **1.30** |

and error model analysis to a selection of functionally diverse input areas (Table 4). The *AICc* analysis showed that the $n_i$ vs $n_s$ relationship is better fitted by the power-law and growth models using the untransformed data for all areas (Table 4), and by a linear model using the log-transformed data (Table 5). In addition, the error model analysis supported a log-normal

**Table 5. ΔAICc for log-transformed data for individual brain areas.**

| dataset | Linear | Quadratic |
|---|---|---|
| VISp | **0.0** | 12.66 |
| VISpm | **0.0** | 16.08 |
| VISl | **0.0** | 14.44 |
| VISam | **0.0** | 14.18 |
| VISal | **0.0** | 16.03 |
| RSPagl | **0.0** | 16.06 |
| RSPd | **0.0** | 15.48 |
| RSPv | **0.0** | 15.79 |
| AM | **0.0** | 15.61 |
| LD | **0.0** | 15.13 |
| LP | **0.0** | 12.13 |
| LGd | **0.0** | 12.35 |
| ORB | **0.0** | 14.71 |
| ACA | **0.0** | 15.89 |
| MOs | **0.0** | 12.31 |
| CLA | **0.0** | 12.24 |
| PTLp | **0.0** | 15.88 |
| TEa | **0.0** | 15.79 |
| AUD | **0.0** | 16.05 |

**Table 6. ΔAICc for residuals distribution for individual brain areas.**

| dataset | log-Normal | Normal |
|---------|-----------|--------|
| VISp | **0.00** | 5.42 |
| VISpm | **0.00** | 28.54 |
| VISl | **0.00** | 6.05 |
| VISam | **0.00** | 12.88 |
| VISal | **0.00** | 17.31 |
| RSPagl | **0.00** | 31.49 |
| RSPd | **0.00** | 26.24 |
| RSPv | **0.00** | 23.66 |
| AM | **0.00** | 9.54 |
| LD | **0.00** | 12.59 |
| LP | 0.82 | **0.00** |
| LGd | **0.00** | **1.16** |
| ORB | **0.00** | 26.22 |
| ACA | **0.00** | 44.72 |
| MOs | **0.00** | 35.05 |
| CLA | **0.00** | 40.35 |
| PTLp | **0.00** | 18.47 |
| TEa | **0.00** | 32.55 |
| AUD | **0.00** | 14.39 |

distribution of residuals (Table 6). We thus fitted the log-transformed $n_i$ vs $n_s$ relationship per input area with a linear model and observed that the resulting fit parameters, y-intercept and slope were correlated and covered a broad range of values (Fig 2A, S5 Fig).

Next we performed simulations using the probabilistic model to try to capture the diversity of input areas by varying systematically the model parameters across a large range. Results from these simulations were log-transformed and fitted with a linear model. The slope vs y-intercept relationship showed similar interaction and range to our data-set (Fig 2B). Further-more, larger values of $N_i$ produced on average higher y-intercept values, while higher values of $p$ were associated with lower slope values (S6 Fig). Varying the width of the input parameter distributions did not affect the mean slope and y-intercept values (S7 Fig).

The connectomes generated by the probabilistic model are unidirectional bipartite net-works and can be analyzed for their network properties, such as their node degree distribution. The in-degree of starter cells ($deg_s$) corresponds to the number of input cells a single starter cell is contacted by, while the out-degree of input cells ($deg_i$) corresponds to the number of starter cells a single input cell contacts (S3 Fig). We decided to investigate how starter and input cell degrees were affected by model parameters, as $n_i$ and $n_s$ varied broadly. Analysis of the degree distributions of both starter and input pools revealed them to vary with both model parameters to various extents (S8 Fig). Next we tried to assess whether manipulating specifi-cally the degree distributions of starter or input cells would affect y-intercept and slope values of the resulting networks. To this end we used a bipartite network configuration model where average degree distributions of both starter and input cells can be declared explicitly (S9 Fig). The resulting $\log(n_i)$ vs $\log(n_s)$ relationship were fitted with a linear model as previously. Changing $deg_s$ affected primarily the y-intercept values, which increased with the mean in-degree of the starter pool, but had little effect on the slope. In contrast, varying $deg_i$ affected primarily the slope, with increasing mean out-degree of the input pool corresponding to lower slope values (Fig 3, S10 Fig). This suggests that the intercept of the $Log(n_i)$ vs $Log(n_s)$ is mainly

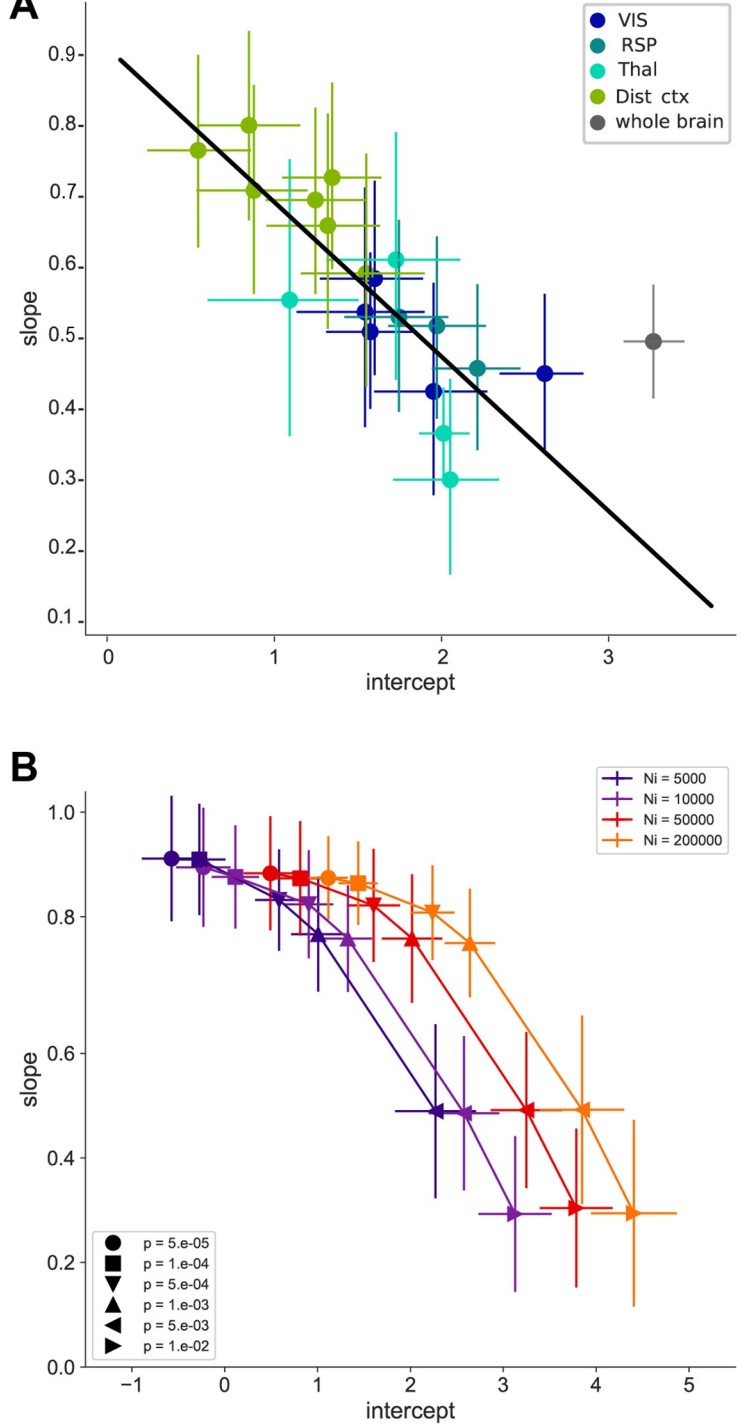

**Fig 2. Distribution of slope and y-intercept values for different input areas.** (A) Slope vs y-intercept relationship for individual input areas. Colors indicate functional grouping of brain areas (VIS (dark blue): VISp, VISpm, VISam, VISl, VISal; RSP (blue): RSPv, RSPd, RSPagl; Thal. (cyan): LP, LD, AM, LGd; Distal cortex (green): ORB, ACA, AUD, PTLp, TEa, MOs, CLA). Whole-brain data is shown in gray. Black line is a linear fit through all data points except the whole-brain data. Error bars are 95% confidence intervals calculated using residuals resampling. (B) Data from simulations using the probabilistic model over a range of parameters ($N_i$, represented by the different colours, and $p$, indicated by different markers).

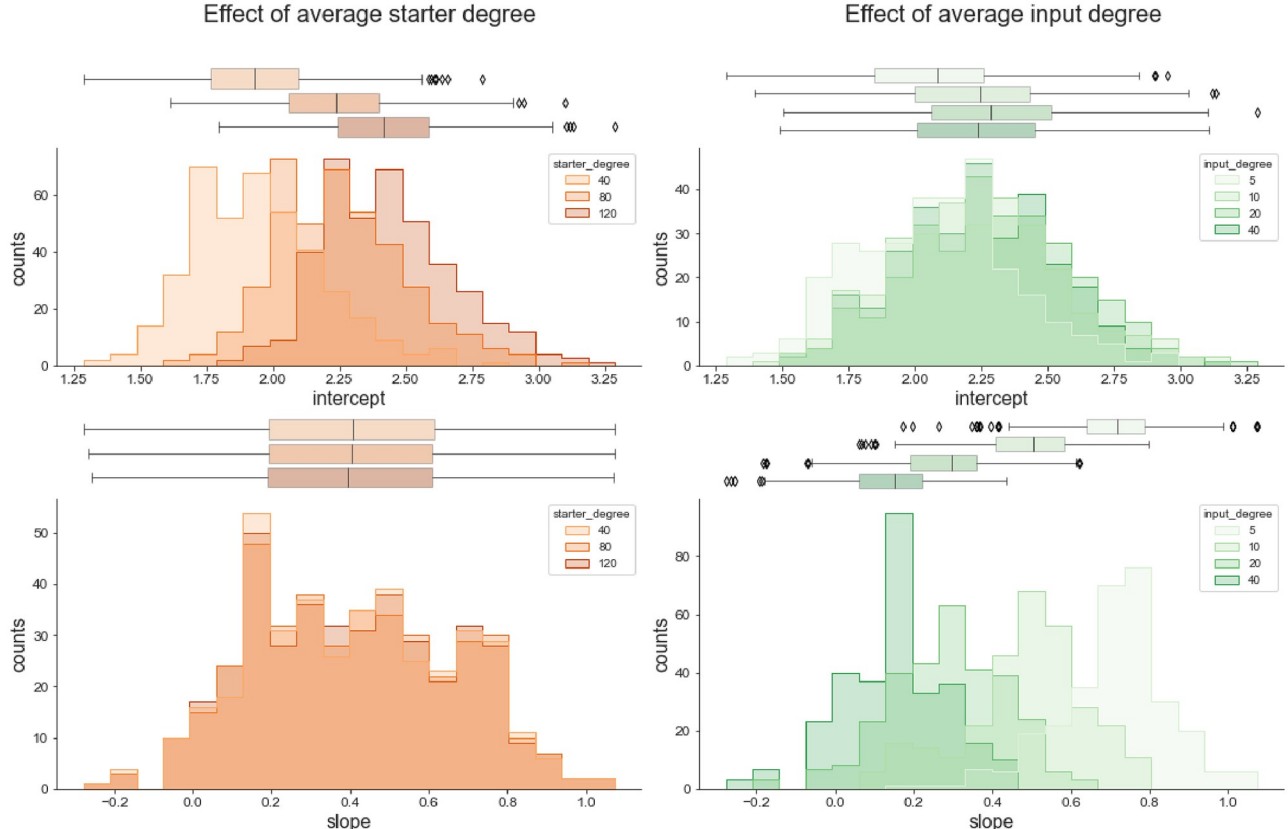

**Fig 3. Effect of degree distribution of input and starter sets on slope and y-intercept values.** Average degree distribution of either the starter (orange) or the input (green) pools were specified using a configuration model and varied separately. $\log(n_i)$ vs $\log(n_s)$ relationships of the resulting networks were fit with a linear model to extract y-intercept and slope values.

affected by the convergence of presynaptic inputs onto single starter neurons, thus the measured y-intercept translates into the number of input cells to a single starter cell. The slope, on the other hand, is mostly determined by the amount of divergence of input cells with respect to the starter cells.

## 2.3 Area input fractions depend on the number of starter cells

Area input fraction (often called "input proportion" or "input fraction" in the literature) is a measure commonly used in rabies tracing studies to reveal input patterns and compare them across experimental conditions such as genetic identity or the location of starter cells populations [14, 15, 19, 22, 25]. It is defined as the number of labelled input cells in a given brain area normalized by $n_i$. This calculation however disregards the range of starter cell numbers and thus implicitly assumes that area input fractions scale with $n_s$ in a uniform fashion across all input areas, an assumption which remains to be tested directly. When $n_s$ is known, another measure called convergence index (also called connectivity ratio or connectivity strength index) is also often reported in the literature. This is calculated by normalising area-wise input cell numbers by $n_s$ and thus it can only provide an $n_s$ independent measure if individual starters cells do not share any of their input cells, a biologically highly unlikely scenario.

Multiple previous studies used multivariate linear regression to evaluate the relative contribution of various experimental parameters on input fraction patterns [15, 19]. We thus used

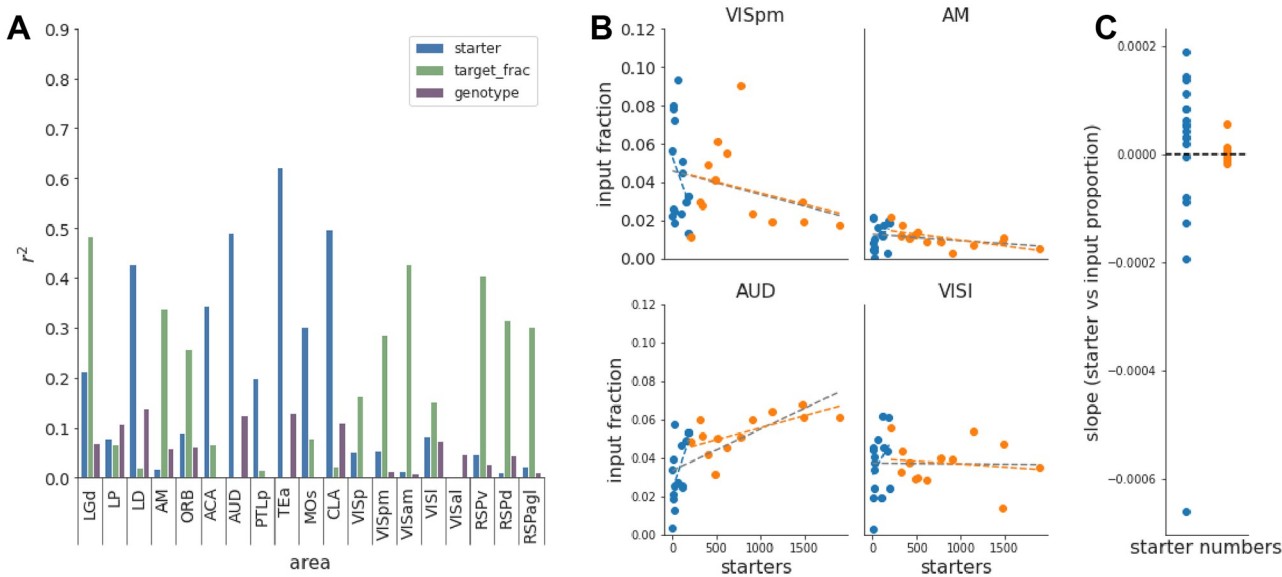

**Fig 4. Effect of starter number on area input fraction.** (A) Multivariate linear regression between the area input fraction and starter cell number, location or genotype of starter cells. (B) Area input fraction vs starter relationship for four example areas. Dashed lines are linear fits through the data, for the full dataset (grey line), starters < 200 (blue line) or starters > 200 (orange line) (C) Slope of the area input fraction vs $n_s$ relationship for low (blue) or high starter numbers (orange).

linear regression analysis on area input fractions and included starter cell number as one of the parameters, either as a single predictor or in combination with other regression variables (the genotype or the location of starter cells). We found that the best predictor of area input fraction variations was the starter cell number for many input areas (Fig 4A, S11 Fig).

Given the prominence of the number of starter cells as a predictor of area input fractions, we next explored its interaction across the range of starter cells in our dataset. We plotted the area input fraction as a function of starter cell numbers for each input area (Fig 4B, S12A Fig). These plots reveal that area input fractions scale with $n_s$ in a highly diverse fashion across areas, with area input fraction vs $n_s$ relationships being either increasing or decreasing for low $n_s$ values, and become asymptotically horizontal for high $n_s$ values. Furthermore, the area input fraction vs $n_s$ relationships have a wider spread for low $n_s$ values(Fig 4B and 4C, S12A Fig). Input areas where $n_s$ was a weak predictor of area input fraction in the regression model (e.g. VISl, RSPd, AM) display an accordingly steady area input fraction vs $n_s$ relationship across the full range of starter cells (Fig 4A, S12A Fig).

We aimed to estimate the number of starter cells beyond which one can consider the horizontal asymptote of the area input fraction vs $n_s$ relationship to have been reached, and thus resulting in $n_s$ invariant area input fraction values. To determine this threshold, we looked for a structural break in the area input fraction vs $n_s$ curves, which would manifest itself as linear regressions on each side of the breakpoint having significantly different slopes [37]. We tested multiple break point values and found that 200 starter cells was the lowest value beyond which the number of areas showing a statistically significant difference for slopes reached its minimum (S12B Fig). Furthermore, the distributions of area-wise slopes from the linear regressions below or above the 200 starter cell cut-off were significantly different (p = 0.03, paired t-test). Importantly, slope values were close to zero for starter cell numbers above 200 (Fig 4C), confirming that in this range, the input fraction does not depend on $n_s$.

We have also examined the relationship of the convergence index to $n_s$. It displayed higher variability for low $n_s$ values and asymptotic convergence at high $n_s$ values (S13 Fig), similarly to area input fraction values. In addition, although all area convergence index vs $n_s$ curves are initially decreasing, the absolute convergence index values for low $n_s$ have more apparent inter-area variability (S14B Fig). This is because the absolute convergence index value is dependent on the area size and not influenced by the relative growth of other brain areas.

To further illustrate how the range of sampled $n_s$ can hinder comparisons of input maps by affecting input patterns and variability, we compared area input fractions calculated for a low and a high $n_s$ group with equal number of observations. Statistical differences in area input fractions between the low- and high starter groups were found in 5 out of the 19 input areas analyzed (S14A Fig). In addition, data from the low starter group showed a much larger variability than the high starter group. Therefore, comparisons using the area input fraction measure should only be used for data sets where starter cell numbers are above the breakpoint value for which the area input fraction vs $n_s$ relationship reaches its horizontal asymptote. Convergence index suffers even more drastically from larger variability in the low-starter group (S14B Fig) and using it to compare datasets with different $n_s$ is inappropriate by construction.

## 2.4 Estimating the number of inputs for a single starter cell

It follows from the analysis above that area input fractions calculated from experiments using a population of starter cells cannot be interpreted at the level of individual starter neurons. However, the y-intercept of the $\log(n_i)$ vs $\log(n_s)$ relationship (Table 7) represents, once converted to linear scale, the number of input cells per starter cell. Using this procedure for

**Table 7. Intercept values from linear fits of $\log(n_i)$ vs $\log(n_s)$ relationships.**

| area | all targets | | VISp | | VISpm | |
|---|---|---|---|---|---|---|
| | mean | CI95 | mean | CI95 | mean | CI95 |
| whole-brain | 3.27 | (3.09, 3.45) | 3.28 | (3.06, 3.49) | 3.23 | (2.75, 3.70) |
| VISp | 2.62 | (2.35, 2.85) | 2.66 | (2.36, 2.91) | 2.32 | (1.60, 2.93) |
| VISpm | 1.96 | (1.60, 2.28) | 1.88 | (1.53, 2.23) | 2.28 | (1.25, 3.03) |
| VISl | 1.60 | (1.27, 1.90) | 1.37 | (0.85, 1.83) | 2.00 | (1.62, 2.36) |
| VISam | 1.55 | (1.14, 1.90) | 1.38 | (0.83, 1.93) | 2.01 | (1.42, 2.49) |
| VISal | 1.58 | (1.31, 1.83) | 1.47 | (1.07, 1.85) | 1.49 | (1.10, 1.94) |
| RSPagl | 1.75 | (1.42, 2.04) | 1.71 | (1.32, 2.10) | 1.86 | (1.16, 2.50) |
| RSPd | 1.98 | (1.68, 2.27) | 1.99 | (1.63, 2.38) | 2.20 | (1.54, 2.77) |
| RSPv | 2.22 | (1.94, 2.47) | 2.24 | (1.92, 2.57) | 2.40 | (1.81, 2.94) |
| AM | 1.10 | (0.60, 1.50) | 0.90 | (0.21, 1.52) | 1.71 | (1.28, 2.14) |
| LD | 2.01 | (1.87, 2.17) | 2.07 | (1.89, 2.28) | 2.16 | (1.88, 2.43) |
| LP | 1.73 | (1.31, 2.11) | 1.96 | (1.50, 2.40) | 2.47 | (1.85, 2.93) |
| LGd | 2.05 | (1.71, 2.35) | 2.10 | (1.88, 2.32) | 1.46 | (0.60, 2.22) |
| ORB | 1.55 | (1.16, 1.90) | 1.46 | (1.03, 1.89) | 1.81 | (0.65, 2.62) |
| ACA | 1.25 | (0.95, 1.55) | 1.12 | (0.76, 1.52) | 1.15 | (0.54, 1.76) |
| MOs | 0.88 | (0.54, 1.20) | 0.80 | (0.34, 1.20) | 0.37 | (-0.34, 1.08) |
| CLA | 0.55 | (0.24, 0.86) | 0.70 | (0.34, 1.08) | 0.71 | (0.04, 1.30) |
| PTLp | 1.32 | (0.95, 1.63) | 1.13 | (0.61, 1.61) | 1.47 | (0.65, 2.14) |
| TEa | 0.85 | (0.54, 1.16) | 0.80 | (0.37, 1.25) | 0.61 | (0.05, 1.26) |
| AUD | 1.35 | (1.05, 1.64) | 1.30 | (0.76, 1.65) | 1.28 | (0.80, 1.86) |

individual input areas, we can then calculate the area input fractions for a single starter cell (S15 Fig). The difference between the two measures is further highlighted by ranking the input areas according to their input fraction (S15C Fig).

Furthermore, as the y-intercept of the $\log(n_i)$ vs $\log(n_s)$ relationship is independent of the starter cell number range, it provides an adequate measure to compare different starter cell populations. To illustrate this, we calculated area input fractions for starter areas VISp and VISpm using the whole range of starter cells (S16A Fig) or using only experiments with starter numbers above 200 (S16B Fig). These measures then can be compared to the y-intercept method (S16C Fig). Differences apparent in the whole starter range comparison (S16A Fig) are driven by unequal slopes across input areas and as such are likely to be misleading. Consequently, a comparison using high starter numbers shows no area to be statistically different between VISp and VISpm (S16A Fig), whereas when using all starters area AM shows a statistically significant difference between VISp and VISpm. Comparisons using y-intercept values however revealed 2 significantly different areas (AM, VISl). The three methods thus provide markedly different results and we argue that only the y-intercept based approach can be interpreted at the level of single starter cells.

## 2.5 Relative connection probability determines the behaviour of area input fractions at low starter numbers

In order to directly assess how connectivity parameters affect the behaviour of the area input fraction vs $n_s$ relationship, we simulated a network with 5 input areas using the probabilistic model introduced in S3 Fig. Across the input areas, we either varied only the connection probability $p$ (Fig 5), only the input pool size $N_i$ (S17 Fig), or both (S18 Fig).

When only $p$ was varied between input areas, the area input fraction varied with $n_s$, as in the experimental dataset (Fig 5A–5D). The influence of $p$ on area input fractions vs $n_s$ is more apparent for low starter cell numbers: areas with the highest connection probability are over-represented in terms of their relative input proportion, while areas with lowest connection probabilities are under-represented (Fig 5D and 5E). For high $n_s$, there is little dependence of area input fractions on $p$, since the area input fraction vs $n_s$ relationship has reached its horizontal asymptote (Fig 5D and 5F). This can be summarized by the decline of the relative input fraction between high and low $n_s$ as a function of normalized $p$ (Fig 5G). In contrast, when only $N_i$ was varied across input areas, the area input fraction for each area was constant across the starter cell range (S17 Fig). Consequently there was no difference in area input fractions between low and high $n_s$ datasets (S17C and S17F Fig). When both $N_i$ and $p$ were varied, the interaction remained essentially the same (S18 Fig), i.e. $p$ drove the difference between low and high $n_s$ datasets.

Finally, in order to obtain an estimate of $N_i$ and $p$ for input areas in our dataset, we fit the data using the likelihood function of the probabilistic model used for simulations. The fitting procedure was performed using maximum likelihood estimation, and returns fitted parameter values for $N_i$ and $p$ (Table 8). Using these values, we can then perform simulations using the same range of starter cells as the experimental data, and observe a good correspondence between experimental and simulated values (Fig 6A, S19 Fig). The plot of area input fractions according to starter cell numbers (low or high starters, respectively Fig 6B top and bottom) reveals very similar trends in fitted and experimental datasets. Given the strong link between relative $p$ and relative area input fraction between high and low $n_s$ (Fig 5E and 5F), the results of this simulation show that the relative value of $p$ per area gets well captured by the multi-area fit of the model (Figs 6B cf. 5G).

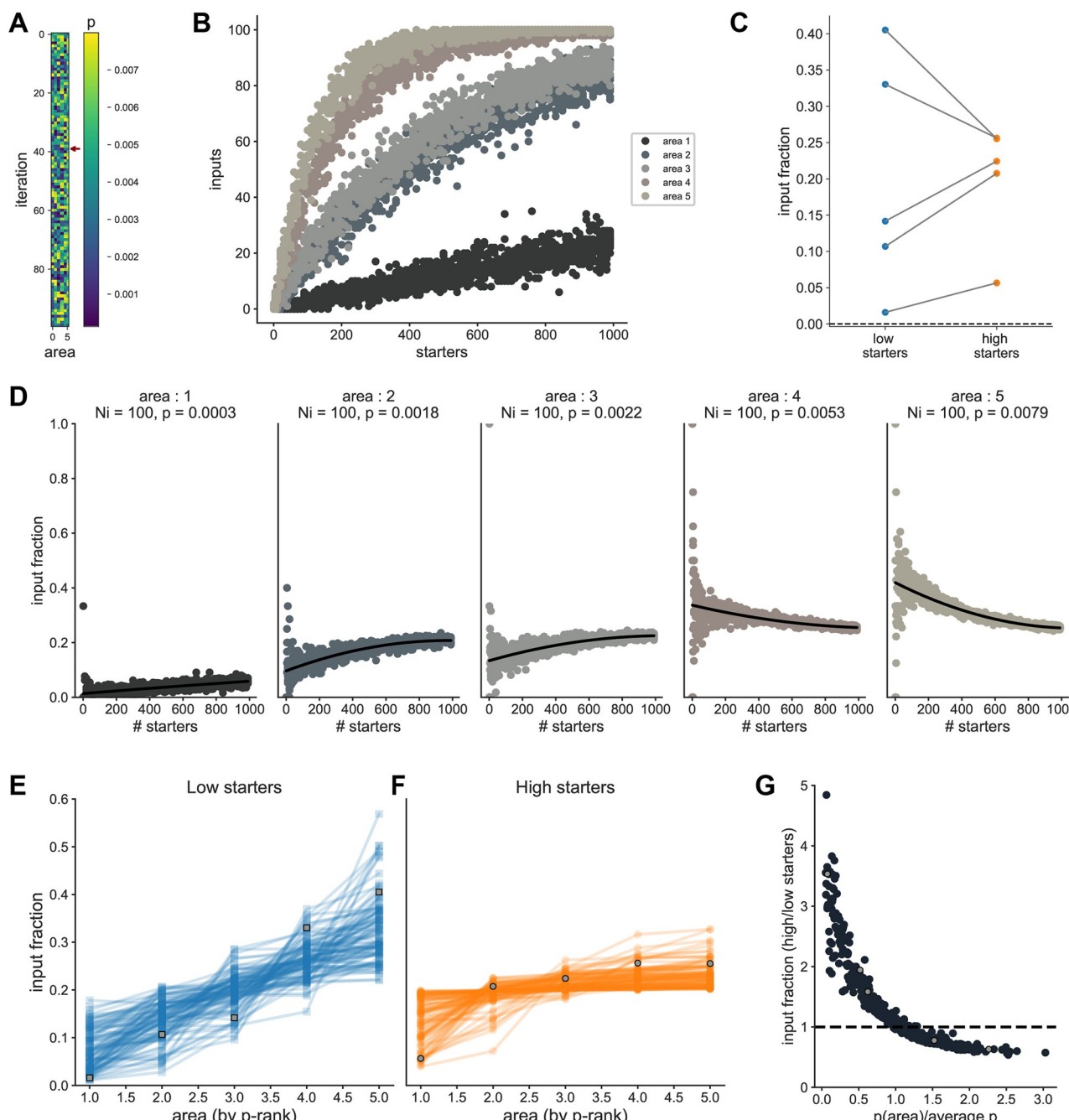

**Fig 5. Effect on relative connection probability on area input fraction vs starter relationship.** Using the probabilistic model, we simulated 5 input areas, all with $N_i$ = 100. 100 independent simulations were repeated to assess the effect of $p$. For each simulation, the connection probability $p_i$ for each input area was randomly drawn between $10^{-4}$ and $8*10^{-3}$. (A) Heatmap showing the combination of connection probabilities used for each simulation. The simulation shown in plots B-D is indicated by a red arrow. (B) $n_i$ vs. $n_s$ relationship for all input areas for one example simulation. (C) Area input fraction for low starter numbers (lowest 10%) or for high starter numbers (highest 10%). (D) Area input fraction vs $n_s$ relationship for each area. Black line is a polynomial fit. (E-F) Area input fraction vs rank of connection probability for low starters (E) or high starters (F). Data from the simulation plotted in B-D are shown in grey. (G) Relationship between the ratio of the area input fraction for high vs low starters, and the normalized connection probability per area. Data from the simulation plotted in B-E are shown in grey.

**Table 8. Parameters obtained from fitting the probabilistic model to experimental data (from Fig 6).**

| area | $N_i$ | | $p$ | |
|---|---|---|---|---|
| | mean | s.d. | mean | s.d. |
| VISp | 3.63e+04 | 3.75e+04 | 5.83e-04 | 9.39e-05 |
| VISpm | 1.01e+05 | 1.28e+04 | 2.99e-05 | 7.73e-06 |
| VISl | 2.17e+04 | 2.37e+04 | 3.51e-04 | 2.41e-04 |
| VISam | 4.80e+04 | 2.09e+04 | 5.46e-05 | 3.30e-05 |
| VISal | 1.32e+04 | 1.56e+04 | 3.56e-04 | 2.13e-04 |
| RSPagl | 1.22e+05 | 3.09e+04 | 2.90e-05 | 1.13e-05 |
| RSPd | 1.40e+05 | 5.32e+04 | 4.05e-05 | 1.66e-05 |
| RSPv | 1.98e+05 | 6.24e+04 | 3.40e-05 | 1.57e-05 |
| AM | 1.26e+04 | 3.27e+03 | 7.48e-05 | 2.91e-05 |
| LD | 8.54e+03 | 1.20e+04 | 4.86e-04 | 2.02e-04 |
| LP | 1.51e+04 | 1.30e+04 | 4.91e-04 | 1.81e-04 |
| LGd | 6.07e+03 | 5.57e+03 | 4.72e-04 | 1.92e-04 |
| ORB | 1.13e+05 | 4.44e+04 | 3.13e-05 | 1.37e-05 |
| ACA | 1.50e+05 | 6.80e+04 | 2.37e-05 | 1.53e-05 |
| MOs | 2.68e+05 | 2.16e+05 | 8.26e-06 | 5.26e-06 |
| CLA | 1.45e+04 | 8.05e+03 | 7.61e-05 | 4.43e-05 |
| PTLp | 3.15e+04 | 2.25e+04 | 1.13e-04 | 5.44e-05 |
| TEa | 4.05e+04 | 3.39e+04 | 9.20e-05 | 6.43e-05 |
| AUD | 5.23e+04 | 6.54e+04 | 1.87e-04 | 1.09e-04 |
| rest | 6.90e+06 | 7.09e+06 | 7.41e-06 | 7.18e-06 |

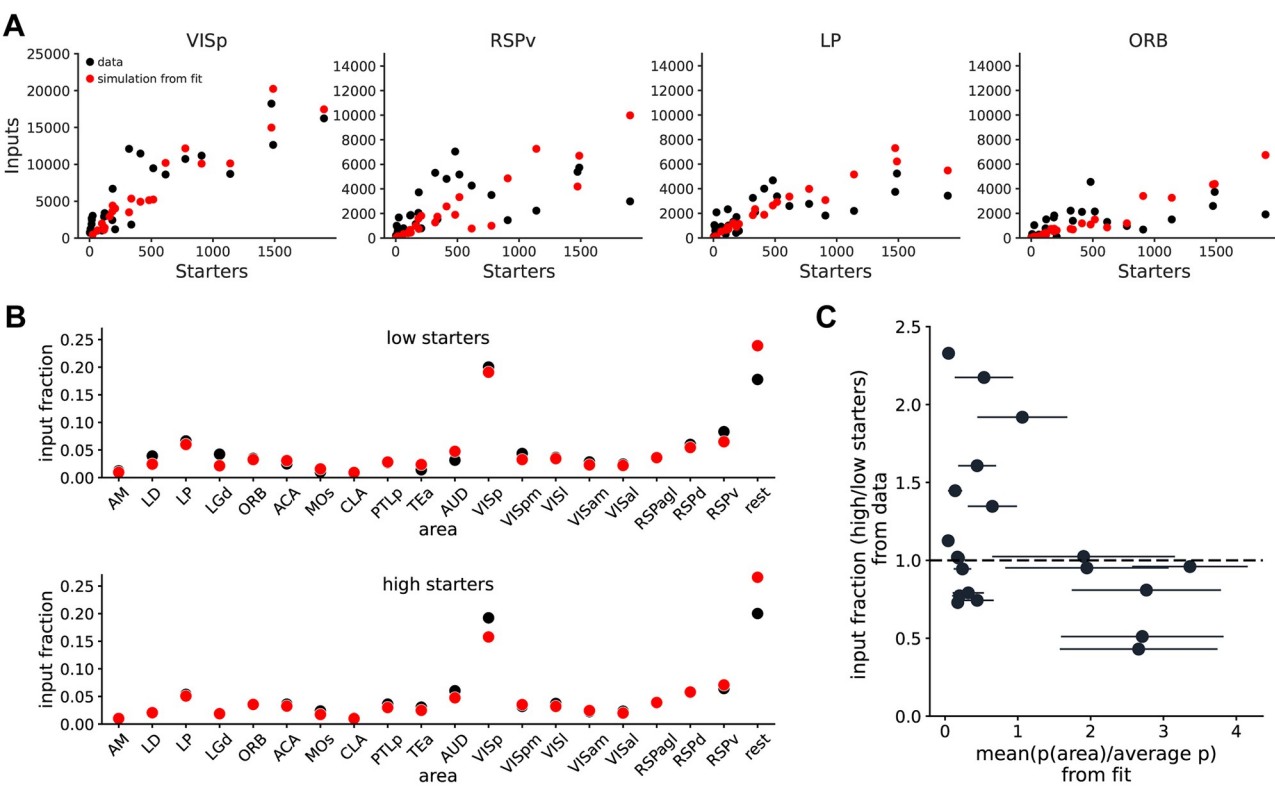

**Fig 6. Estimation of $N_i$ and $p$ per area in experimental dataset.** (A) Four example areas with input vs starters relationships for the data (black) or simulations with parameters obtained from the model fit of the data (red), for one iteration of the fit. (B) Plot of experimental input fraction ratio between high and low starters vs relative $p$ across areas obtained from fitting values.

## 3 Discussion

### 3.1 Experimental caveats

There are several technical caveats concerning the labelling and detection of neurons. Regarding labelling, we have used two helper viruses: one expressing the TVA-receptor and EGFP, and one expressing the G-protein but no fluorophore. The modified rabies virus was expressing mCherry. Neurons co-expressing EGFP and mCherry were thus defined as starter cells. However, this may result in neurons counted as starter cells but deficient in G protein, thus leading to an over-estimation of starter cells. Conversely, G protein only expressing neurons could become retrogradely infected through local connections and propagate rabies labelling to a second layer of monosynaptically connected neurons, thus leading to an overestimation of local input cells. We attempted to mitigate these confounds by injecting a mixture of high titre ($\sim 10^{14}$ genome-copies/ml) helper viruses thus optimising for co-infection [38]. Regarding labelling by rabies virus, it is unknown if all connections have the same probability of propagation or not and our method is unable to distinguish between connection and propagation probability. Trans-synaptic spread depends on the presence of rabies virus receptors [39], is modulated by activity [40] and likely by the number and size of synaptic contacts between given input and starter cells. Future work is required on virus strains with enhanced neurotropism, on the identity of rabies receptors and on quantitative evaluation using orthogonal methods such as dense anatomical reconstruction allowing unequivocal identification of synaptic connections.

### 3.2 Cell counting

Some of the external datasets analysed in S1 Fig were quantified by counting labelled cells in separate brain slices. This approach can introduce bias and be a further source of variability. While advances in design-based stereological methods have improved cell number estimations from tissue slices [41], they typically rely on uniform cell densities and are rarely applied when quantifying rabies tracing data. To illustrate the problem, we have run numerical simulations using a range of biologically relevant cell densities (see methods). Counting cells in every slice or every 2nd, 3rd or 4th slice consistently resulted in varying degrees of systematic overestimation (S20 Fig). Furthermore, the variance between individual experiments was considerable, especially toward low cell densities. To further illustrate the problem using real data, we have virtually sectioned two brains from our dataset and counted cells in every 4th slice. The estimated whole brain cell counts varied considerably depending on slice sampling. Importantly, estimates greatly differed from the cell counts obtained by whole-tissue 3D cell counting (Table 9). These results argue against estimating cell numbers by counting cells in sectioned tissue, especially when only sampling some slices. The imprecision introduced may underlie some of the model selection uncertainties in S1 Fig. To achieve unbiased, accurate cell counts, cell counting in the whole 3D volume of interest is essential.

While automated 3D cell counting can deal with very large datasets and offers consistent results, systematic cell detection biases which scale with the label density cannot be excluded (e.g. under-detection at input areas with extremely high cell densities). However, this is

**Table 9. 2D detection artefacts.** Number of inputs from 2 brains using 3D detection or for simulated 2D detection from consecutive 50 $\mu$m wide slices, keeping every 4th slice, assuming cell radii of 10 $\mu$m.

| true number of inputs | slice 1 of 4 | slice 2 of 4 | slice 3 of 4 | slice 4 of 4 |
|:---:|:---:|:---:|:---:|:---:|
| 49479 | 45068 | 66880 | 96336 | 64680 |
| 7785 | 5500 | 16480 | 15060 | 6576 |

unlikely to manifest in our results as model selection analysis is consistent across input areas with different label densities. In addition, datasets from multiple laboratories and acquired through different imaging and cell counting methods show the same relationship. Interestingly, a dataset generated with no explicit cell counting, but rather quantifying projection volume by counting pixels reaching a certain fluorescence threshold (Allen Institute), shows a different residuals distribution (normal). Yet the $n_i$ vs $n_s$ relationship is still best described by a power-law relationship. The difference in residuals distribution analysis for this dataset may be expected through a presumably sub-linear increase and saturation in the projection volume with high cell densities. Additionally, an over-estimation for areas of low cell density is similarly expected, as the inclusion of neuropil has a disproportional effect at low cell densities.

Our results generalize well across external datasets, but it is important to note that the starter areas in all datasets explored here are cortical (except [32, 34], which are from the olfactory bulb and amygdala, respectively). It remains to be determined how the $n_i$ vs $n_s$ relationship generalises to profoundly different neuronal structures, such as the cerebellum, striatum or the spinal cord.

### 3.3 Model selection and error structure

Metrics used in previous studies implicitly assumed linear scaling between the number of input and starter cells. We performed model selection analysis on our data (on whole brain inputs or area-defined inputs) and a range of external datasets, to compare the linear relationship with a range of candidate models that could describe the increase in number of inputs cells with increasing number of starter cells. For most datasets, the models with the lowest AICc values were the power-law and growth models (Table 1). This model selection analysis is limited by the data (both the number of observations, and the range of starter cells) and by the list of candidate models provided. For instance, although our dataset does not present any obvious saturation of the number of labelled inputs, one would expect a saturating regime to be reached if enough starter cells were labeled. Therefore this analysis does not aim at identifying a "true" underlying model, but at defining how to fit the data for consistent descriptive analysis. However, it does show that the linear model is consistently outperformed by competing non-linear models.

In addition, it is apparent that the $n_i$ vs $n_s$ relationships show a large variance in the data, and a skewed residuals distribution. The error structure analysis supports log-transformation of the data, which results in more normally distributed residuals (Fig 1B and 1D). We thus performed further analysis after log-transformation.

Model selection on log-transformed data shows that a linear model provides a good description of the datasets. Although this doesn't represent the underlying model in log scale, it appears the range of $n_s$ sampled corresponds to a linear part of the log-transformed dataset. We thus used a linear model for descriptive fitting.

### 3.4 Do rabies strains impact extracted parameters?

In recent years, viral tracing tools have considerably improved, thanks to the introduction of engineered glycoproteins [42] and more efficient rabies strains [6] that increased efficiency of synaptic transfer and reduced neuronal toxicity. One concern would be that using different rabies strains could lead to different $n_i$ vs $n_s$ relationship models. We analysed a large number of datasets comprising data from two different rabies strains (SAD-B19 or CVS-N2c), a large range of starter cell types and various starter areas. For the overwhelming majority of the datasets, our model selection analysis returned the same qualitative distribution (Tables 1–3 and S1 Fig). We however did observe quantitative differences in $\log(n_i)$ vs $\log(n_s)$ relationships

depending on the rabies strain used, with a higher intercept in the fit of the $\log(n_i)$ vs $\log(n_s)$ relationships for CVS-N2c. This is unsurprising given the improved retrograde transfer observed with this virus strain [6]. If the analysis introduced here is used to deduce connectivity parameters, or to compare them across different datasets, one should be mindful of the potential influence of the rabies strains used.

### 3.5 Effect of starter cell range on area input maps

In order to describe input maps obtained from rabies tracing experiments, inputs are typically either normalized to the total number of input cells in the same brain (area input fraction) or the number of starter cells (convergence index). We show directly that neither of these measures are independent of $n_s$ (S12 and S13 Figs), and make comparisons between starter populations unreliable (S14 Fig). The large and systematic biases are driven by two factors: first, the large variability of both measures at low starter cell numbers; and second, the different slopes of $n_s$ vs area input fraction or convergence index for low and high $n_s$. Thus averaging either measure across the full range of $n_s$ results in averaging across different behaviours. These biases are not surprising since neighbouring starter cells are likely to have some shared input cells.

The steepest non-linearity between area input fraction and number of starter cells is observed for small numbers of starter cells, before becoming close to constant for large numbers (Fig 4, S12A Fig). Using data from experiments with $n_s$ over which area input fractions are stable does mitigate this problem, yet the biological meaning of such derived area input fractions are far from trivial. As area input fractions are driven by the ensemble of different input areas, quantitative comparisons between populations with different input areas cannot be interpreted. However, these experiments do offer descriptive power and valuable qualitative insight of the census of input areas. For populations that share input areas to a large extent (e.g. starter populations with largely overlapping dendritic fields), quantitative comparison can be valid. In this case, area input fractions in a range independent of $n_s$ show the relative propensity of a given input area to contact starter cells in the target region. This can be particularly useful not only to compare different but intermingled starter populations, but especially to compare the connectivity of the same starter population across e.g. treatments or time [40, 43, 44].

If the high $n_s$ data can be averaged to lead to meaningful area input fractions values under certain conditions, what can be inferred from low $n_s$ experiments? We observe that while convergence index vs $n_s$ relationships are always decreasing before reaching their constant value, input fraction vs $n_s$ relationships depend on the relative growth of the area inputs with respect to total brain input, and can be therefore either increasing or decreasing in the low range of $n_s$ values (Fig 4C). This behaviour of area input fraction vs $n_s$ relationships for low $n_s$ informs on the growth of input area fraction and thus the connectivity probability ($p$) of input areas relative to each other (Fig 5).

### 3.6 Biological meaning of parameters

Systematic exploration of the parameter space of the probabilistic model, followed by fitting the resulting $\log(n_i)$ vs $\log(n_s)$ relationships, revealed the relationships between y-intercept, slope, and the connectivity parameters of the model. First, we observed that for a given value of $p$, the mean intercept increases with $N_i$, while the slope varies little. In contrast, varying $p$ for a given value of $N_i$ affects both the slope and intercept. Furthermore, increasing the average degree of the starter set (i.e. the number of connections received by individual starter cells) leads to a shift of the mean intercept towards higher values, with no effect on the slope. Increasing the average degree of the input set (i.e. the number of connections given by an

individual input cell) on the other hand leads to a decrease in the mean slope value. This is consistent with the shifts in distributions of both starter and input degrees observed when varying $N_i$ and $p$ using the probabilistic model (S8 Fig). Using these observations we can attach a biological meaning to some of these parameters.

The measured y-intercept translates into the number of input cells to a single starter cell—a measure otherwise only obtainable by single-cell initiated rabies tracing [1, 2]. Fitting our data-set yields a y-intercept of 3.2, thus we estimate $\sim$1860 rabies labelled input neurons per starter cell. While the efficacy and specificity of transsynaptic rabies transmission present significant unknowns discussed elsewhere [43], this is not a wholly unreasonable estimate given a recent study reporting 7500 synapses for L2/3 pyramidal cells [45] and assuming multiple synapses per connection. The proportional contribution of different input areas to a single starter cell can be derived from the area-wise y-intercept measures, but it is important to keep in mind that there is no trivial mapping between rabies labelling and functional input strength [43, 46].

The fit values for input pool size ($N_i$) represent the size of the population of presynaptic cells which can be connected to the starter cell population. Assuming individual input neurons to carry somewhat independent information, $N_i$ informs about the possible input diversity arriving from a given area. Based on this analysis, MOs, ACA, RSPv and RSPd give the most diverse inputs to L5PNs across VISp and VISpm. Conversely, as the input pools of e.g. VISal, AM and LGd are small, these connections are likely to have less information capacity.

### 3.7 Conclusion and recommendations

When planning rabies tracing experiments, one should first consider the required conclusions to be drawn. In general, starter cells should always be counted and a significantly larger number of samples, in the order of tens, should be planned. Starter cell numbers should ideally cover a broad range (from single digits to thousands), with denser sampling in the low $n_s$. In addition, the starter cell population should be as homogeneous as possible. Experimental data thus obtained can be subjected to model selection, and used to estimate biologically pertinent parameters, such as y-intercept and $N_i$. If only a small number of experiments are feasible, one should aim for a large number of starter cells where area input fractions have less variability across experiments and are close to a steady state. Area input fractions can in this case be used to compare similar starter cell populations e.g. before and after treatment. However, this approach requires estimating the starter cell number range where area input fractions are stable, which is likely to be dependent on multiple factors, including starter cell type and rabies strain used.

Tracing synaptic connectivity using modified rabies viruses is a powerful method in the toolkit of neuronal cartography, especially so when interpreted correctly. Just like with any other experimental method, methodically different approaches are essential to validate the conclusions and results should not be taken as ground truth. On top of these general concerns, the number of starter cells needs to be considered carefully both when planning experiments and when analysing rabies-obtained input maps.

## 4 Methods

### 4.1 Animals and viruses

All animal experiments were prospectively approved by the local ethics panel of the Francis Crick Institute (previously National Institute for Medical Research) and the UK Home Office under the Animals (Scientific Procedures) Act 1986 (PPL: 70/8935). The following transgenic mice on a C57BL/6 background were used: Tg(Colgalt2-Cre)NF107Gsat (RRID: MMRRC_036504-UCD, also known as Glt25d2-Cre); Tg(Rbp4-Cre)KL100Gsat/Mmucd

(RRID: MMRRC_031125-UCD); Tg(Tlx3-cre)PL56Gsat (RRID: MMRRC_041158-UCD). Animals were housed in individually ventilated cages under a 12 hr light/dark cycle.

EnvA-CVS-N2cΔG-mCherry rabies virus, and adeno-associated viruses expressing TVA and EGFP (AAV8-EF1a-flex-GT) or CVS-N2c glycoprotein (AAV1-Syn-flex-H2B-N2CG) were a generous gift from Molly Strom and Troy Margrie. The AAV-EF1a-Cre plasmid (Plasmid #55636) and retrograde AAV2-retro helper vector (Plasmid #81070) were purchased from Addgene and generously packaged by Raquel Yustos and Prof. Bill Wisden at Imperial College London.

## 4.2 Surgical procedures

Surgeries were performed on mice aged 5–12 weeks using aseptic technique under isoflurane (2–3%) anaesthesia, and analgesia (meloxicam 2 mg/kg and buprenorphine 0.1 mg/kg). The animals were head-fixed in a stereotaxic frame and a small hole (0.5–0.7 mm in diameter) was drilled in the skull above the injection site. The virus solution was loaded into a glass microinjection pipette (pulled to a tip diameter of 20 $\mu$m) and pressure injected into the target region at a rate of 0.4 nl/s using a Nanoject III delivery system (Drummond Scientific). To reduce backflow, the pipette was left in the brain for approximately 5 min after completion of each injection.

For rabies virus tracing experiments using cre driver lines a 1:2 mixture of TVA and CVS-N2c glycoprotein expressing cre-dependent AAVs (5–20 nL) was injected at stereotaxic coordinates (VISpm: lambda point—0.8 mm, ML 1.6 mm, DV 0.6 mm; VISp: lambda point—1.0 mm, ML 2.5 mm, DV 0.6 mm). For the TRIO experiments, 50 nL AAVretro-Ef1a-Cre was injected into LP (lambda point—1.7 mm, ML 1.5 mm, DV 2.4 mm) followed in 3–8 weeks by the injection of helper AAVs as described above. Rabies virus (50–100 nL) was injected 5–7 days later at the same site. Ten to twelve days later, animals were transcardially perfused under terminal anaesthesia with cold phosphate-buffer (PB, 0.1 M) followed by 4% paraformaldehyde (PFA) in PB (0.1 M).

## 4.3 Data acquisition and analysis

Brain samples were embedded in 4–5% agarose (Sigma-Aldrich: 9012–36-6) in 0.1M PB and imaged using serial two-photon tomography [47, 48]. Eight optical sections were imaged every 5 $\mu$m with 1.2 $\mu$m x 1.2 $\mu$m lateral resolution, after which a 40 $\mu$m physical section was removed using a vibrating blade. Excitation was provided by a pulsed femto-second laser at 800 or 930 nm wavelength (MaiTai eHP, Spectraphysics). Images were acquired through a 16X, 0.8 NA objective (Nikon MRP07220) in three channels (green, red, blue) using photomultiplier tubes. Image tiles for each channel and optical plane were stitched together using the open-source MATLAB package StitchIt (https://github.com/SainsburyWellcomeCentre/StitchIt). For 3D cell detection the open-source package *cellfinder* [28] was used. Registration to the Allen CCFv3 [27])and segmentation was done using the *brainreg* package [17]. Cell coordinates were downsampled to 10 $\mu$m to match the resolution of the Allen CCFv3 space and the number of cells was counted for each segmented area. To reduce the occurrence of false positives, the hindbrain areas (HB) were removed from the whole brain cell counts, as these areas are known not to project directly to the neocortex.

Area-wise cell count data were imported into Python 3.6 and all further analysis performed using custom scripts. In the following Methods sections, we refer to $n_s$ as the number of starter cells, and $n_i$ as the number of input cells, or $n_{i, A}$ for the number of input cells in area $A$. For fitting relationships, the explanatory variable $x$ can refer to $n_s$ or log($n_s$), and the dependent variable $y$ can refer to $n_i$ or log($n_i$).

Cell counts for all experiments and the code used for analysis and generating figures can be found at https://github.com/ranczlab/Tran_Van_Minh_et_al_2023.

## 4.4 Descriptive statistics and model selection

The relationship between number of input cells and number of starter cells was fitted with multiple models using the *lmfit* package. The of candidate models used for model selection are: linear model, quadratic model, power-law model, exponential model, and a growth model defined by the equation:

$$y = \frac{y_{max} * x}{k + x}$$

For each model the Akaike Information Coefficient corected for small sample size ($AIC_c$) can be calculated as:

$$AIC_c = 2k + n \log\left(\frac{RSS}{n}\right) + \frac{2k(k+1)}{n - k - 1}$$

where $n$ is the sample size, $k$ is the number of parameters of the model. The $AIC_c$ for different models were compared, and if the difference in $AIC_c$ values for two models is larger than 2, the model with the lowest $AIC_c$ is considered to have better support [49].

## 4.5 Analysis of residuals distribution

If the residuals from a fit of the untransformed dataset follow a normal distribution, further analysis should be performed the untransformed dataset. If residuals follow a log-normal distribution, it is more appropriate to perform further analysis on log-transformed data. We used the method described in [31] to determine the error structure of the dataset. Briefly, we calculated the likelihood that the data is generated from a normal distribution with additive error:

$$\mathcal{L}_{norm} = \prod_{i=1}^{n} \left[ \frac{1}{\sqrt{2\pi\sigma_{NLR}^2}} \exp\left( \frac{-(y_i - a_{NLR} x_i^{b_{NLR}})^2}{2\sigma_{NLR}^2} \right) \right]$$

and the likelihood that the data is generated from a lognormal distribution with multiplicative error:

$$\mathcal{L}_{lognorm} = \prod_{i=1}^{n} \left[ \frac{1}{\sqrt{2\pi\sigma_{LR}^2}} \exp\left( \frac{-(\log(y_i) - log(a_{LR} x_i^{b_{LR}}))^2}{2\sigma_{LR}^2} \right) \right]$$

where $n$ is the sample size. The *AICc* for each error model was then calculated as:

$$AIC_c = 2k + n \log(\mathcal{L}) + \frac{2k(k+1)}{n - k - 1}$$

and the two models with the lowest $AIC_c$ value considered as having better statistical support if the difference in $AIC_c$ values is larger than 2.

## 4.6 Estimation of fit parameters

In order to estimate the distribution of fit parameters for the $\log(n_i)$ ($y$) vs $\log(n_s)$ ($x$) relationships, we used residuals resampling. Briefly, for each bootstrap iteration i, a linear model was fit to the data to obtain fitted values $\hat{y}_i$ and residual values $\epsilon_i$. The new values to fit were obtained by adding to a randomly resampled residual value from the initial fit ($y_{B,i} = \hat{y}_i + \epsilon_{i,j}$).

The linear model was fit to this iteration. This step was repeated 5000 times and the resulting fits analyzed to obtain the confidence intervals of the fit parameters.

### 4.7 Probabilistic model

We modeled unidirectional connections between input areas $I_a$ of size $N_{i_a}$ and a starter area $S$ of unspecified size. $p_{I_a}$ is the probability that a neuron in area $I$ is connected to a neuron in area $S$. For each observation, we sample $N_s$ cells from the starter area, and build for each input area the adjacency matrix of size $(N_{i_a}, N_s)$ that represents the connections between all $N_{i_a}$ neurons and all $N_s$ neurons. Each element $m_{i,j}$ of the adjacency matrix can take as values 0, if the i-th neuron of area $I_a$ is not connected to the j-th neuron of area $S$, or 1 if the neurons of this pairs are connected, with $P(m_{i,j} = 1) = p_I$. Rabies tracing experiments were simulated by building an adjacency matrix per observation of the inputs vs starter graph, and repeating these observations over a similar range of starter numbers as a typical rabies experiment. Brain-to-brain variability was represented by sampling, for each observation, the model parameters from discretized normal distributions, truncated to keep only positive numbers, with specified mean and standard deviation.

### 4.8 Effect of the number of starter cells on area input fraction

Area input fractions were defined as the ratio between the number of cells in each input area in ipsilateral side, and the total inputs counted in the same brain. We use a Chow test [37] to test for structural breaks in the slope of the area input fraction vs starters relationships.

Multivariate linear regression analysis was used to compare the relative effect of starter cell numbers, starter cell locations, and starter cell genotype, on area input fractions for all regions of interest. The regression models were defined using the *ols* function from the *statsmodel* Python package. Models with different combinations of the predictors were also assessed. Statistical significance of the models were assessed by an F test and all *p* values corrected for multiple comparisons using the Benjamini-Hochberg method. The predictors used were the starter cells number, genotype, and their location, represented by the target frac parameter. Because of the close proximity of the targeted injection locations (VISp and VISpm) for infection of starter cells, some brains had starter cells in both targeted areas. We scaled the ratio of starter cells in either area to represent the continuum between brains for which all starter cells are in VISp (corresponding to a target frac value of 1) and brains where all starter cells are in VISpm (target frac value of -1).

### 4.9 Bipartite configuration model

The adjacency matrices generated by the simulations define bipartite graphs, made of two sets of respectively $N_i$ and $N_s$ nodes. The degree of a node *n* corresponds to the number of edges connecting this node to the rest of the network. In order to assess the effect of the degree of each set on the fit parameters of the $N_i$ vs $N_s$ relationship, we built graphs of specified degrees distributions using the *bipartite_configuration_model* generative model function from the *networkx* package.

The degree distribution of the starter set was sampled from a normal distribution centered around its chosen average degree. Degrees of the corresponding input set are then picked iteratively from a normal distribution centered around its chosen average s degree, so that the sum of starter and input degrees are equal.

## 4.10 Analysis of the relative effects of connection probability and input pool sizes on area input fraction

In order to assess the effect of each model parameter on area input fractions, we used the probabilistic model to simulate 5 input areas and varied parameters independently. To assess the effect of connection probability, $p_{I_a}$ for each area was randomly drawn between $10^{-4}$ and $8*10^{-3}$, and $N_{i_a}$ set to 100. To assess the effect of input area size, $N_{i_a}$ was a discrete value randomly picked between 100 and 500, and all $p_{I_a}$ were set at $5*10^{-4}$. We also considered the effects of varying both $p_{I_a}$ and $N_{i_a}$ within the same ranges. Simulations were performed for up to $N_s$ = 1000 starter cells. We refer to data with $N_s$ in the bottom 10% and top 10% of this range as low starters group and high starters group, respectively. For each input area, the area input fraction vs starter relationship was fit with a second degree polynomial equation.

## 4.11 Derivation of the probabilistic model and fit of experimental data

For the probabilistic model for a single input area of size $N_I$ and connection probability $p_I$, the likelihood of observing $n_i$ inputs for $N_s$ sampled starters can be formulated as:

$$\mathcal{L}(N_I, p_I) = \frac{N_I!}{n_i!(N_I - n_i)!}\left((1 - (1 - p_I)^{N_s})^{n_i}\right)(1 - p_I)^{N_s(N_I - n_i)}$$

or, for an ensemble of areas:

$$\mathcal{L} = \prod_a \frac{N_{I_a}!}{n_{i,a}!(N_{I_a} - n_{i,a})!}\left((1 - (1 - p_{I_a})^{N_s})^{n_{i,a}}\right)(1 - p_{I_a})^{N_s(N_{I_a} - n_{i,a})}$$

where $N_{I_a}$ and $p_{I_a}$ are the area the input pool size and connection probability of area $a$, respectively.

This equation was used to fit the experimental data, using the *differential_evolution* algorithm from the *scipy* package for global optimization.

Bounds passed to the fitting function were calculated as follows. First, area volumes were downloaded from the Allen Mouse CCF (volumes at 25 $\mu$m isotropic resolution) using the Allen SDK. Number of neurons per area were deduced using neuronal cells densities per brain areas from [50, 51]. If the density for a specific area was not specified in [50], we used the density for the next level up in the Allen hierarchy as a proxy. For $N_i$, lower and upper bounds were defined as 1/40 and 1/(1.5) of the number of estimated neurons per area, rounded up to the nearest thousand, respectively. For $p$, bounds were 1E-07 and 6.00E-04, respectively, for all areas expect the "rest of brain" area, for which bounds were 1.00E-08 and 1.00E-04. Bounds for all areas are listed in Table 10. The fit was performed 100 times with different seed values to assess the effect of initialization parameters.

The resulting $N_{I_a}$ and $p_{I_a}$ values for each area were then used with the probabilistic model to obtain the values plotted in Fig 6.

## 4.12 Classification of rabies strains across datasets

We used all datasets with starter cell identified as Pyramidal cells or interneurons and input cells quantified as cell counts. Classification was performed using the *sklearn* package. Rabies strain was used as the target of the classification, and cell type, starter numbers and input numbers used as features. Starter and input numbers were normalized using *MinMaxScaler*. Linear support vector classification was performed using the *LinearSVC* classifier. We used stratified k-fold cross-validation with k = 4. Briefly, the data was randomly divided into k subsets, so

**Table 10. Bounds per area for fitting the probabilistic model to experimental data (from Fig 6).**

| area | $N_i$ | | $p$ | |
|---|---|---|---|---|
| | min | max | min | max |
| VISp | 2.70E+04 | 7.33E+05 | 1.00E-07 | 6.00E-04 |
| VISpm | 4.00E+03 | 1.08E+05 | 1.00E-07 | 6.00E-04 |
| VISl | 5.00E+03 | 1.27E+05 | 1.00E-07 | 6.00E-04 |
| VISam | 3.00E+03 | 8.20E+04 | 1.00E-07 | 6.00E-04 |
| VISal | 3.00E+03 | 7.80E+04 | 1.00E-07 | 6.00E-04 |
| RSPagl | 6.00E+03 | 1.54E+05 | 1.00E-07 | 6.00E-04 |
| RSPd | 9.00E+03 | 2.48E+05 | 1.00E-07 | 6.00E-04 |
| RSPv | 1.10E+04 | 2.85E+05 | 1.00E-07 | 6.00E-04 |
| AM | 1.00E+03 | 1.60E+04 | 1.00E-07 | 6.00E-04 |
| LD | 3.00E+03 | 6.80E+04 | 1.00E-07 | 6.00E-04 |
| LP | 3.00E+03 | 8.00E+04 | 1.00E-07 | 6.00E-04 |
| LGd | 1.00E+03 | 3.50E+04 | 1.00E-07 | 6.00E-04 |
| ORB | 7.00E+03 | 1.89E+05 | 1.00E-07 | 6.00E-04 |
| ACA | 1.10E+04 | 2.83E+05 | 1.00E-07 | 6.00E-04 |
| MOs | 4.40E+04 | 1.19E+06 | 1.00E-07 | 6.00E-04 |
| CLA | 1.00E+03 | 3.70E+04 | 1.050E-07 | 6.00E-04 |
| PTLp | 5.00E+03 | 1.25E+05 | 1.00E-07 | 6.00E-04 |
| TEa | 7.00E+03 | 1.91E+05 | 1.00E-07 | 6.00E-04 |
| AUD | 1.60E+04 | 4.24E+05 | 1.00E-07 | 6.00E-04 |
| rest | 5.00E+05 | 1.00E+08 | 1.00E-08 | 1.00E-04 |

that the created folds preserve the distribution of classes observed in the complete dataset. Each subset was iteratively used as the test set while the other k-1 subsets were used for training. The resulting k classification scores were averaged to obtain a final score.

### 4.13 Cell counting artefacts in brain sections

First, we created 0.5 x 0.5 x 4 mm tissue blocks and randomly populated them with cells of 20 $\mu$m diameter in densities varying from 10 to 40000 cells / mm$^3$. Next, overlapping cells were removed, resulting in a final density range of 10 to 20000 cells / mm$^3$, the maximum corresponding to roughly 20% of total neuronal densities in mouse cortex [51]. We then simulated slicing the tissue block into 50 and 100 $\mu$m thick slices and counted cells in every slice, or every 2nd, 3rd or 4th slice and multiplied the counts accordingly. We repeated these experiments 100 times and calculated the average ratio between the real cell numbers in the tissue block and the estimates.

### Supporting information

**S1 Fig. Model comparison for whole-brain data for other datasets.** Colours indicate different fitted models, with the same colour-code as in Fig 1. Datasets are from [13–15, 18, 19, 21, 22, 25, 32, 36]. Please note that input quantification is done by counting labelled pixels instead of individual neurons in the Allen Institute dataset (panel O).
(TIF)

**S2 Fig. Classification of rabies strain in pooled datasets.** (A) Classification of the rabies strain used based on $\log(n_s)$, $\log(n_i)$ and starter cell type, for all pooled datasets where starter

cell type was clearly identified as either pyramidal cells or interneurons and inputs quantified as cell counts. The model was a linear support vector classifier, and we used stratified 4-fold cross-validation to preserve the percentage of samples for each class. The plot corresponds to a single cross-validation fold. (B), Cross-validation scores show consistent accuracy across folds.
(TIF)

**S3 Fig. Probabilistic model.** Illustration of input parameters, iteration steps and output measures for the probabilistic model.
(TIF)

**S4 Fig. Variability in connectivity parameters leads to skewness of residuals.** (A) Distributions of connectivity parameters ($p$, average $5*10^{-4}$, and $N_i$, average 50000 cells; both distributions have a s.d. of 0.2 $*$ their average value). (B) Simulation of inter area connectivity with the probabilistic model plotted as in Fig 1. Connectivity parameters are randomly drawn from the distributions in A for each observation. (C) Simulations as in B were performed 100 times and residual analysis was performed for each resulting curve. Dotted line represents dAICc of -2.
(TIF)

**S5 Fig. Linear fits per input area.** Linear fits of log-transformed $n_i$ vs $n_s$ relationship for individual brain areas.
(TIF)

**S6 Fig. Distributions of fit parameters for $\log(n_i)$ vs $\log(n_s)$ relationships with varying $N_i$ and $p$.** Distributions of slope (A) and y-intercept (B) values obtained across simulations with various model parameters (colours for $N_i$ and shading for $p$), as plotted in Fig 2. Both $N_i$ and $p$ were drawn from distributions with a width of 0.2 $*$ average.
(TIF)

**S7 Fig. Varying the width of model parameter distributions has little effect on fit parameters of $\log(n_i)$ vs $\log(n_s)$ relationship.** (A) Distributions of fit parameters of $\log(n_i)$ vs $\log(n_s)$ relationship for an average $N_i$ = 10000, varying connection probabilities as in Fig 2 and parameters drawn from distribution of varying widths (S.D. = 0.1, 0.2 or 0.4 $*$ average). (B) Slope vs y-intercept plot for an average $N_i$ = 10000 with both model parameters drawn from distribution of varying widths.
(TIF)

**S8 Fig. Degree distributions from simulations with the probabilistic model.** (Top) Distributions of starter cell degrees for varying $N_i$ and $p$ parameters. Both parameters were drawn from distributions of with 0.2 $*$ average value of parameter. (Bottom) Distributions of input cell degrees for varying $N_i$ and $p$ parameters. Both parameters were drawn from distributions of with 0.2 $*$ average value of parameter.
(TIF)

**S9 Fig. Configuration model.** Illustration of a single step of the simulations for the configuration model.
(TIF)

**S10 Fig. Intercept vs slope relationships in simulations using the configuration model.** (Top) Influence of starter degree on intercept vs slope relationships (each panel is a specified mean input degree). (Bottom) Influence of input degree on intercept vs slope relationships (each panel is a specified mean starter degree).
(TIF)

**S11 Fig. Multivariate linear regression of the area input fraction across areas using combined predictors.**
(TIF)

**S12 Fig. Relationship between area input maps and $n_s$ for individual brain areas.** (A) Input fraction vs $n_s$. Dashed lines represent linear fit through all data (grey), for $n_s < 200$ (blue) or $> 200$ (orange). (B) p-value for Chow-test for varying break point values (x-axis), for individual brain areas.
(TIF)

**S13 Fig. Relationship between area convergence index and $n_s$ for individual brain areas.** Convergence index vs $n_s$, for $n_s < 200$ (blue) or $> 200$ (orange).
(TIF)

**S14 Fig. Input maps for low or high starter cell number.** (A) Area input fractions averaged across the low starter range (<125 starters, n = 10, blue) or across the high starter range (>600 starters, n = 10, orange). Statistical differences between area input fraction for low and high $n_s$ are indicated by *. Significance was calculated using multiple t-tests corrected for multiple comparisons using the Benjamini-Hochberg method with a false discovery rate of 10%. (B) Same as A, using convergence index per area.
(TIF)

**S15 Fig. Area input fraction vs input fraction per cell for input areas.** (A) Area input fraction calculated over the full range of starter cells (error bars are s.d.) or (B) calculated from the y-intercept of $\log(n_i)$ vs $\log(n_s)$ relationship converted to linear scale (error bars are 95% confidence intervals from residuals bootstrap). (C) Areas ranks obtained via both methods are showed as a heatmap (lowest rank correspond to smallest fraction, lighter colors).
(TIF)

**S16 Fig. Using area input fraction or y-intercept to compare experimental parameters.** (A) Area input fraction calculated over the full range of starter cells, for experiments with target area in VISp (dark blue) or VISpm (light blue). Asterisks indicate significant difference. Significance is calculated using multiple t-tests and is corrected for multiple comparisons using the Benjamini-Hochberg method with a false discovery rate of 10%. (B) Same as A, but for $n_s > 200$. (C) Y-intercept of log-transformed $n_i$ vs $n_s$ relationship. Significance is assessed by subtracting bootstrapped values of the y-intercepts between target areas. If the resulting distribution does not contain 0, the intercepts are considered significantly different. NB: areas VISp and VISpm act either as local or distal input areas, depending on the starter cells' location.
(TIF)

**S17 Fig. Varying relative size of input pools has no effect on area input fraction vs starter relationship.** Simulations performed using the probabilistic model with 5 input areas (100 iterations). For each iteration, the connection probability for each input area was $p = 5*10^{-4}$ and the size of the number of input cells $N_i$ per area was randomly drawn between 100 and 500. (A-F), as in Fig 5.
(TIF)

**S18 Fig. Effect of varying relative size of input pools and connection probability on area input fraction vs starter relationship.** Simulations performed using the probabilistic model with 5 input areas (100 iterations). For each simulation, the connection probability $p$ for each input area was randomly drawn between $1*10^{-4}$ and $8*10^{-3}$ and the size of the number of

input cells $N_i$ per area was randomly drawn between 100 and 500. (A-F), as in Fig 5, (G-H), same as (E-F) but to compare the effect of relative $N_i$.
(TIF)

**S19 Fig. Simulation of input vs starters relationships using $N_i$ and $p$ obtained from fit of experimental dataset.** Input vs starters relationships for the data (black) or simulations with parameters obtained from the model fit of the data (red), for one iteration of the fit, for all areas not shown in Fig 6.
(TIF)

**S20 Fig. Simulation of counting cells in physically sliced tissue.** Top: the ratio between cell numbers estimated by counting in sliced tissue and the number of cells in the volume plotted versus cell density. Colours indicate slice sampling. Simulation data for two slice thicknesses, 50 and 100 $\mu$m, are shown. Bottom: variance of cell counts versus cell density for different slice sampling values and slice thickness.
(TIF)

**S1 Text.**
(TXT)

## Acknowledgments

We thank Troy Margrie and Molly Strom for viral constructs, Rob Campbell, Charlie Rousseau and Adam Tyson for help with data acquisition and analysis of rabies tracing experiments, Gavin Kelly for help with model selection, Marco Beato, Jonny Kohl and Petr Znamenskiy for comments on the manuscript, and all colleagues who shared their data in S1 Fig. For the purpose of Open Access, the author has applied a CC BY public copyright licence to any Author Accepted Manuscript version arising from this submission.

## Author Contributions

**Conceptualization:** Alexandra Tran-Van-Minh, Ede Rancz.

**Data curation:** Alexandra Tran-Van-Minh, Zhiwen Ye, Ede Rancz.

**Formal analysis:** Alexandra Tran-Van-Minh, Zhiwen Ye, Ede Rancz.

**Funding acquisition:** Ede Rancz.

**Investigation:** Zhiwen Ye.

**Methodology:** Alexandra Tran-Van-Minh.

**Project administration:** Ede Rancz.

**Supervision:** Ede Rancz.

**Visualization:** Alexandra Tran-Van-Minh.

**Writing – original draft:** Alexandra Tran-Van-Minh, Ede Rancz.

**Writing – review & editing:** Alexandra Tran-Van-Minh, Ede Rancz.

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
