## [Decision Letter · Decision Letter 0]

18 Oct 2022

PONE-D-22-21238Quantitative analysis of rabies virus-based synaptic connectivity tracingPLOS ONE

Dear Dr. Rancz,

Your manuscript is an important and noteworthy contribution to the field. Your submission will be accepted for publication very rapidly once a few very minor edits are completed. Please carefully review the comments provided by Reviewer 1. Is is not necessary to respond to all of the comments, but several of their suggestions will improve the clarity of the manuscript. Reviewer 1's comment to define the biological meaning of model parameters/metrics earlier in the text will improve readability. For example, the editor agrees with the reviewer's comment that the meaning/interpretation of the slope/intercept is unclear until the end of the section. Providing the biological interpretation of these metrics earlier will be clearer. While the manuscript is very well written overall, the editor notes at least one instance of a grammatical error: “We find that the relationship between number of the starter cells and number of the labelled input cells in non-linear”. I think it should read “is non-linear”. There are a few minor errors noted, such as in the color coding description in Figure 2 (orange not blue). Please carefully edit the manuscript, as PLOS One does not provide any copyediting services after acceptance.

We look forward to receiving your revised manuscript.

Kind regards,

Ian B Hogue, Ph.D.

Academic Editor

PLOS ONE

Journal Requirements:

“This work was supported by the Wellcome Trust (104285/B/14/Z) and the Francis Crick Institute, which receives its core funding from Cancer Research UK (FC001153), the United Kingdom Medical Research Council (FC001153), and the Wellcome Trust (FC001153). A.TVM. received funding from the European Union’s Horizon 2020 research and innovation programme under the Marie Sklodowska-Curie grant agreement No 747902.”

“This work was supported by the Wellcome Trust (104285/B/14/Z) and the Francis Crick Institute, which receives its core funding from Cancer Research UK (FC001153), the United Kingdom Medical Research Council (FC001153), and the Wellcome Trust (FC001153). A.TVM. received funding from the European Union’s Horizon 2020 research and innovation programme under the Marie Sklodowska-Curie grant agreement No 747902. The funders had no role in study design, data collection and analysis, decision to publish, or preparation of the manuscript.”

Reviewers' comments:

Reviewer's Responses to Questions

**Comments to the Author**

1. Is the manuscript technically sound, and do the data support the conclusions?

Reviewer #1: Yes

2. Has the statistical analysis been performed appropriately and rigorously? 

Reviewer #1: Yes

3. Have the authors made all data underlying the findings in their manuscript fully available?

Reviewer #1: Yes

4. Is the manuscript presented in an intelligible fashion and written in standard English?

Reviewer #1: Yes

5. Review Comments to the Author

Reviewer #1: The manuscript by Tran-Van-Minh et al. assesses the parameters that affect data quantification in experiments using monosynaptic rabies tracing due to unstandardized metrics. Through this study, they identified that the relationship between the number of starter cells (ns) and input cells (ni) is not linear but logarithmic transformation of ns an ni can allow them to behave linearly. The use of the slope and intercept can inform the relationship between starter and input cells. They identify that starter cell number is a vital parameter and offer great suggestions in the quantification of rabies tracing experiments. I believe this study is necessary and timely, and will benefit the field in setting metrics for the quantification of such experiments.

Concerns:

1) Figure 1, citing the other distributions, can be made in this figure. This figure doesn’t show the residual plot of log-transformed data for quadratic equation fit. However, no citing for the respective fit. (The residual plots are present in the supplemental).

a. Good fitness of the curves selected should be cited from table 1.

2) Figure 2, the description of the biological meaning of the intercept and the slope is not clear in the paper. The authors mention these in the last paragraphs of the manuscript, but a upfront description and interpretation will greatly benefit the readability.

3) Figure 3, color description is inaccurate (orange not blue). The meaning of the degrees across the paper is not well described.

4) Parameters describing the FS2 are not apparent in the evaluation for clustering.

5) In general, the description and meaning of parameters are not clear (such as p) are not clear across the paper.

6. PLOS authors have the option to publish the peer review history of their article (what does this mean?). If published, this will include your full peer review and any attached files.

Reviewer #1: No

---

## [Author Response · Author response to Decision Letter 0]

1 Nov 2022

We are grateful for the feedback and have now addressed all questions and suggestions, as detailed below. We believe the suggested changes made for a stronger manuscript and we are looking forward to your decision. 

1) Figure 1, citing the other distributions, can be made in this figure. 

Done. 

This figure doesn’t show the residual plot of log-transformed data for quadratic equation fit. However, no citing for the respective fit. (The residual plots are present in the supplemental).

Done. 

a. Good fitness of the curves selected should be cited from table 1.

Table 1 is cited in the relevant text passage.

2) Figure 2, the description of the biological meaning of the intercept and the slope is not clear in the paper. The authors mention these in the last paragraphs of the manuscript, but a upfront description and interpretation will greatly benefit the readability.

The biological interpretation has now been added to the end of section 2.2.

3) Figure 3, color description is inaccurate (orange not blue). The meaning of the degrees across the paper is not well described.

This has been corrected now. 

4) Parameters describing the FS2 are not apparent in the evaluation for clustering.

The methods section (4.12) contains the relevant details. 

5) In general, the description and meaning of parameters are not clear (such as p) are not clear across the paper.

We now mention the interpretation of interpretable parameters at several points in the manuscript. However, we are careful not to overinterpret the parameters which are no pliable to a simple biological meanings.

---

## [Editor Report · Decision Letter 1]

9 Nov 2022

Quantitative analysis of rabies virus-based synaptic connectivity tracing

PONE-D-22-21238R1

Dear Dr. Rancz,

We’re pleased to inform you that your manuscript has been judged scientifically suitable for publication and will be formally accepted for publication once it meets all outstanding technical requirements.

Kind regards,

Ian B Hogue, Ph.D.

Academic Editor

PLOS ONE
---

## [Editor Report · Acceptance letter]

4 Jan 2023

PONE-D-22-21238R1 

Quantitative analysis of rabies virus-based synaptic connectivity tracing 

Dear Dr. Rancz:

I'm pleased to inform you that your manuscript has been deemed suitable for publication in PLOS ONE. Congratulations! Your manuscript is now with our production department. 

Kind regards, 

on behalf of

Dr. Ian B Hogue 

Academic Editor

PLOS ONE